# Regulators and Conductors of Immunity: Natural Immune System in Health and Autoimmunity

**DOI:** 10.3390/ijms26115413

**Published:** 2025-06-05

**Authors:** Katalin Böröcz, Dávid Szinger, Diána Simon, Timea Berki, Péter Németh

**Affiliations:** 1Department of Immunology and Biotechnology, Clinical Center, Medical School, University of Pécs, 7624 Pécs, Hungary; 2Medical School, University of Pécs, 7624 Pécs, Hungary

**Keywords:** natural immune system, natural autoantibodies (nAAbs), immune function, autoimmunity, systemic lupus erythematosus, rheumatoid arthritis, systemic sclerosis, type 1 diabetes, innate immunity, adaptive immunity, immunological tolerance, infectious diseases, pathogenesis, compensatory mechanisms, IgM isotype, therapeutic strategies

## Abstract

Natural autoantibodies (nAAbs) recognize self-antigens and are an important component of the immune system, having evolved from invertebrates to vertebrates, and are viewed as stable byproducts of immune function and essential players in health and disease. Initially characterized by their conserved nature and multi-reactivity, primarily as IgM isotypes, nAAbs are now recognized for their adaptability in response to infections and vaccinations, bridging innate and adaptive immunity. The nAAbs and the cellular elements, such as γδ T, iNKT, and MAIT cells, of the natural immune system perform a primary defense network with moderate antigen-specificity. This comprehensive literature review was conducted to analyze the role of natural autoantibodies (nAAbs) in health and disease. The review focused on research published over the past 40 years, emphasizing studies related to infectious diseases, vaccinations, and autoimmune disorders. Recent studies suggest that nAAbs engage in complex interactions in autoimmune diseases, including systemic lupus erythematosus, rheumatoid arthritis, systemic sclerosis, and type 1 diabetes. Their roles in immunological processes, such as maternal tolerance during pregnancy, further underscore their complexity. Emerging evidence indicates that nAAbs and the cellular elements of the natural immune system may contribute to both disease pathogenesis and protective mechanisms, highlighting their dual nature. Continued research on nAAbs is vital for improving our understanding of immune responses and developing therapeutic strategies for autoimmune disorders and infectious diseases.

## 1. Introduction

When Irun Cohen proposed revising the clonal selection theory and replacing it with the cognitive paradigm, there were already a lot of data on natural autoimmunity [1]. The presence of autoantibodies in the blood serum of healthy individuals without clinical symptoms of autoimmune diseases has long been known about.

The study of natural autoantibodies (nAAbs) has evolved significantly over the past few decades, prompting a deeper exploration of their multifaceted roles in health and disease. Historically viewed as static byproducts of immune function, nAAbs are now recognized for their dynamic contributions to immune regulation, autoimmunity, and vaccination responses. Recent studies focusing on the natural immune system have led to a better explanation of the biological role of the interaction between nAAbs and the cellular components of the immune system.

Specialized subsets of cellular immunity have been described over the last few decades. The evolutionary aspects of some T cell subpopulations with invariant T cell receptor chains were clear, but their role in immune regulation was not known. Similarly, the potential interactions of the gamma–delta (γδ) T, invariant natural killer T cells (iNKTs), and mucosa-associated invariant T cells (MAITs) with the nAAb network and nAAb-producing B1 cells were studied only recently. γδ T cells, iNKTs, and MAITs are subgroups of (unconventional) innate-like T cells, collectively making up ~10–30% of the total T cell compartment [2] and performing many roles the innate immune system, but also phenotypically resembling and having strong ties to cells of the adaptive immunity [2,3]. On the other hand, literature about the connections these innate-like T cells might have with nAAbs and the innate-like B cell subgroup, B1 cells, is scarce.

The present paper aims to help better orient the vast and heterogeneous literature on cellular and humoral components (nAAbs) of the natural immune system, often characterized by contradictory findings, and to clarify their complex role in different immunological contexts. By mapping the scientific advances in nAAb research along a timeline, we aim to highlight gaps in knowledge, guide future investigations, and emphasize the potential of nAAbs as biomarkers and therapeutic targets in autoimmune and infectious diseases.

## 2. Cellular and Humoral Components and Their Role in the Natural Immune System

### 2.1. Potentially Mutual Influence of γδ T Cells and nAAb Network

Gamma-delta (γδ) T cells, which make up about 0.5–10% of circulating lymphocytes, act as a connection between innate and adaptive immunity due to their unique properties [4]. These cells play a significant role in supporting immune defense, tumor surveillance, and B cell maturation, and are involved in autoimmune responses [4]. γδ T cells recognize antigens in their natural form and do not require antigen presentation by the major histocompatibility complex (MHC) [5]. They aid in the maturation of autoreactive immature B cells in the spleen by providing key signals—such as IL-4 production and CD30L interaction—that encourage B cells to develop into antibody-producing cells capable of recognizing a broader range of antigens [4]. In γδ T cell-deficient models, immature B cells were found to stall in their development, which suggests γδ T cells are crucial in maintaining proper B cell maturation, potentially impacting autoimmune processes when absent or dysfunctional [4].

In contrast to most unstimulated αβ T cells, which are antigenically naive and metabolically inactive, many peripheral γδ T cells in non-immunized mice are already in a state of moderate activation [6], and there are clues that γδ T cells continuously affect the nAAb repertoire in mice; thus, they might have a connection to B-1 and marginal zone (MZ) B cells [7]. However, this might be a dynamic bidirectional connection, as nAAbs help opsonize dead or damaged cells and pathogens, marking them for clearance by phagocytes. The γδ T cells may respond to this clearance process, recognizing the cellular stress signals produced during phagocytosis. This process can create a local environment rich in cytokines and chemokines, which may activate γδ T cells, prompting them to produce inflammatory cytokines (e.g., IL-17, IFN-γ) that further modulate the immune response [8,9]. The γδ T cells can also present antigens and release cytokines that modulate both innate and adaptive responses. If γδ T cells recognize stress or microbial signals in an environment where nAAbs have already marked pathogens or apoptotic cells, they may be further activated to recruit more immune cells or enhance the antibody production by B cells [9,10,11]. On the other hand, γδ T cells can help counterbalance potential autoimmune activation by nAAbs against self-antigens via regulatory cytokines (such as IL-10). This regulation could be particularly relevant in preventing excessive inflammation or autoimmunity when nAAbs bind to self-antigens, as seen in chronic inflammatory conditions or autoimmune diseases [12,13,14,15].

### 2.2. Connections of iNKT and B-1 Cells

iNKTs are capable of detecting glycolipids on a non-polymorphic MHC class I-like molecule, CD1d, with a T cell receptor composed of a moderately variable α chain and a mostly germline-defined β chain [16]. Similarly to conventional T cells, iNKTs also have many subgroups, each presenting different and specific functionality [17]. The general function of iNKTs involves mounting an early and rapid immune response to pathogens until the adaptive immune response is deployed [16,17,18], but they also have key regulatory roles in autoimmunity, cancer, infection, and tolerance [19], as well as bystander activation of both CD8+ and CD4+ memory T cells, contributing to immunological memory maintenance [20]. There might be a multi-level interaction between B-1 cells and iNKT, consisting of antigen presentation, cytokine signaling, and nAAb-mediated feedback.

Many cells present CD1d; however, antigen-presenting cells have upregulated expression, and among all, marginal zone (MZ) B cells (CD21^high^ CD23^low^ IgM^high^ IgD^low^) and B-1 cells stand out in the level of expression in mice and humans [21,22,23], marking the potential of MZ- or B-1-mediated antigen presentation to iNKT cells. CD1d cycles from the cell surface back into an endosomal network, where it is loaded for presentation within the same endosomal and lysosomal compartments that process foreign protein antigens [23]. In these compartments, CD1d replaces its glycolipid ligands with either endogenous or externally derived lipids before returning to the cell surface [23].

Considering cytokine interactions, iNKT has a regulatory-supportive role in many B cell subgroups, including autoreactive B cells [24]. In contact sensitivity, a study [25] found that antigens and IL-4 from iNKT activate B-1 cells via the STAT-6 signaling pathway as part of a cooperative interplay between iNKT and B-1 cells.

On the level of autoantibodies, an in vitro study showed that physiological doses of polyclonal IgM inhibit α-gal-ceramide-induced IFN-γ production of iNKT [26,27]. Related studies suggest that activated iNKT plays a role in murine renal ischemia, which IgM nAAbs do not mediate directly, but IgM nAAbs protect against it through the regulation of iNKT, although the exact mechanism is unknown and is only suggested to involve receptor inhibition [27].

### 2.3. Conjecture of MAIT and B-1 Cell Interaction

Both B-1 lymphocytes and mucosal-associated invariant T (MAIT) cells are primarily present around the barriers: B-1 is predominantly in peritoneal and pleural cavities, with a small portion in the lymph nodes, and spleen [28] and MAIT cells are present around the mucosal surface, lungs, liver, and blood, with importance in host defense and tissue repair [29,30,31,32]. MAIT performs innate-like as well as Th1- or Th17-like functions depending on TCR-dependent or -independent activation [30]. MAIT cells express a semi-invariant TCRα chain that recognizes small molecules, such as pterin analogs and riboflavin metabolites, presented by the non-polymorphic MHC class I-related molecule MR1 [33].

MAIT cells can activate B-2 cells and influence antibody production, class-switch, and memory B cell formation [34,35]. A subgroup of MAIT has TCRs cross-reacting with self-structures and is proposed to have regulatory functions in immune homeostasis and involvement in pathological processes [36]. To date, potential B-1 and MAIT cell interplay is a blind spot of the literature, but it remains an exciting conjecture that these cells, on the basis of self-recognition, might interact with nAAb-producing B-1 cells, as we have marked on Figure 1.

### 2.4. B-1 Cells and Natural Autoantibodies (nAAbs)

Human B-1 cells, originating mainly in the fetal liver, are found in the serous layer of body cavities and mucosal tissues, playing a pivotal role in early immune defense. Identified in both mice and humans, B-1 cells are marked by the CD20^+^ CD27^+^ CD43^+^ CD70^−^ profile in humans, contributing to immunity by remarkable phagocytic activity resulting in antigen presentation on both MHC class I and II [37,38,39] and the production of broad-reactivity antibodies, influencing T cell activity and targeting conserved antigens [37,38,39,40,41,42]. B-1 cells can further be divided based on cell surface expression of CD5 (CD5+ B-1a and CD5- B-1b cells) [43]. In comparison to B-1a, cells of the B-1b subset have higher IgM production but milder inducibility by LPS [44,45]. B-1b also has an enhanced CCR6-regulated migration to the spleen [45]. Vergani et al. (2022) [46] performed a time-stamping experiment on mice B cells, resulting in new insights that challenge the mainstream idea that B-1 cells are naïve and Ig production is unbiased by antigens [40,47]. Instead, they proposed that B-1 activation happens in a predominantly neonatal developmental window shared with other antigen-experienced memory B cell compartments prior to the activation of B-1 cells by self-antigens and foreign antigens. Specifically, B-1a cells shall be accounted for as predominantly neonatally induced IgM memory subsets with self-sustaining capabilities, but this scheme can be helpful for studying B-1b, MZ B, iNKT, and MAIT cells as well [46].

Both B-1 cell subsets produce natural antibodies that recognize both protein and non-protein antigens, showing promise for T-independent responses and vaccine development, especially against resistant bacteria [48]. In infections such as Borrelia and SARS-CoV-2, B-1-derived antibodies rapidly target stable antigens, providing adaptive-like immunity and cross-reactivity with unrelated pathogens. This flexibility, unlike the specificity of B-2 antibodies, allows B-1 nAAbs to adapt to evolving pathogens, potentially offering sustained immune coverage and relevance for immunotherapy [37,38,39,40,41,42,48,49,50,51,52,53,54,55,56,57,58,59,60,61].

### 2.5. Role of Natural Autoantibodies (nAAbs)

One way to look at physiological autoimmunity is the so-called immune computation model [62]. The immune system, consisting of innate and adaptive receptors and effectors, processes molecular signals reflecting the body’s condition—such as infection, trauma, malignant transformation, or cellular aging. Based on predefined rules, it computes a molecular- and cellular-level response that leads to outcomes like cell death, proliferation, differentiation, migration, and blood vessel formation, ultimately promoting healing and tissue remodeling or contributing to disease. This process also allows the immune system to adjust its rules for future responses.

Antibodies serve as essential molecular mediators between the organism, its immune system, and both symbiotic (microbiome) and pathogenic foreign entities, functioning at cellular and molecular levels. While the complete functional characterization of the entire antibody repertoire (the “antibodyome”) remains unfinished, immunoglobulins can be classified based on their origin and target (Figure 2). Serological assessments offer an indirect but valuable representation of numerous underlying immunological processes, many of which cannot be fully understood by clinical manifestations alone.

### 2.6. Natural IgM Autoantibodies

Natural antibodies are produced without (known) antigenic stimulation, unlike adaptive antibodies generated by B-2 lymphocytes with T cell help. In humans, B-1-like cells, constituting a large portion of the umbilical cord, and adult circulating B cells resemble the self-reactive B-1 population in mice, suggesting an early immune role [41,63,64]. B-1 cell identity, linked to its B cell receptor (BCR), can be transferred to non-self-reactive B-2 cells through allelic replacement, further demonstrating BCR’s role in B cell function and development [65].

Most natural antibodies are natural autoantibodies (nAAbs) that bind to self-structures like damage-associated molecular patterns (DAMPs), cytoskeletal proteins, and mitochondrial proteins, forming a broad, low-affinity immune network [65,66,67]. nAAbs predominantly belong to the IgM isotype and are polyspecific, recognize repetitive patterns efficiently (making them similar to receptors of the innate immune system, e.g., TLRs) and derive from nearly unmutated germline genes [68,69,70,71].

IgM natural autoantibodies (nAAbs) are polyclonal and polyreactive, targeting conserved self-antigens and pathogen-associated patterns similar to innate immune receptors. Specific IgM nAAbs, like anti-phosphorylcholine, can recognize AB0 antigens, endotoxins, and apoptotic cell markers but not nuclear antigens or IgG, exemplifying that each IgM nAAb has a specific recognition pattern [27,66,72].

These IgM nAAbs play a dual role: they neutralize pathogens and toxins while aiding in the clearance of apoptotic cells, reducing inflammation, and preventing pathogenic IgG autoantibody induction [40,73].

Unlike adaptive antibodies, IgM nAAb levels do not necessarily rise with autoantigen levels, allowing for continuous production by B-1 cells independent of antigen exposure [66]. Some IgM nAAbs can bind to and neutralize self-reactive IgG, mitigating autoimmune risk, although certain conditions—like hepatitis C infection—can lead to the formation of immune complexes that may cause kidney and skin complications [26,67,71,74,75,76,77,78,79,80,81].

### 2.7. The Dual Nature of Natural IgG Autoantibodies: Implications for Immune Tolerance and Autoimmune Disease Development

IgG natural autoantibodies (IgG nAAbs) exist in normal serum but are often masked by anti-idiotypic IgM nAAbs. Like IgM nAAbs, IgG nAAbs can bind conserved self-antigens, with around 15–20% of murine IgG showing polyreactivity, primarily in the IgG3 isotype produced by B-1 cells [71,79,82,83,84]. IgG nAAbs become active in mice after exposure to gut bacteria and in humans after around two years, with infections further increasing their levels [85,86,87,88,89,90].

More than a decade ago (2013), it was hypothesized that natural IgG autoantibodies (nAAbs) are abundant in human serum, with individual profiles that were stable over time but varied by age, gender, and disease, suggesting links to central tolerance and autoimmune risk [90]. This inspired further research to test the stability of nAAb profiles and their transition to pathological autoantibodies (pAAbs) in autoimmune conditions. Studies in NZB mice, a model for autoimmune hemolytic anemia, showed that nAAb levels against conserved antigens rise with age before disease onset, at which point pAAb levels increase, indicating plasticity in the nAAb pool [86,91,92,93,94,95,96,97,98].

Although under certain conditions (e.g., genetic predisposition [99,100,101] or repeated immunization [102]) B-1 cell-derived nAAbs can serve as templates for the development of higher-affinity, class-switched pathological autoantibodies (particularly those in the intersection of pathological autoantibodies and nAb sets in Figure 2), their exact physiological and pathogenic roles have yet to be fully elucidated [70].

Intravenous immunoglobulin (IVIG) therapy, rich in IgG nAAbs, is in use for autoimmune neuropathies, systemic immune-mediated conditions, pediatric autoimmune diseases, and dermatological autoimmune conditions with diverse efficacy, as recently reviewed by Giovanna Danieli et al. (2025) [103]. IVIG is a crucial therapy in autoimmune neuropathies, but the off-label use in individual indication provides accumulating data about its effectiveness in systemic autoimmune conditions, especially with skin involvement.

As nicely reviewed by Schwab and Nimmerjahn (2013) [104], a multitude of in vitro studies group the potential underlying mechanism of action of IVIG into F(ab’)_2_-dependent pathways or Fc-dependent routes.

IVIG antibodies can bind with their F(ab’)_2_ region to idiotypic determinants of pAAbs, effectively neutralizing their pathogenic effect (restoring anti-idiotype network) or exerting an anti-inflammatory effect by neutralizing pro-inflammatory cytokines (anti-cytokine nAAbs), blocking cellular receptors, or labeling effector cells for deletion by antibody-dependent cellular cytotoxicity. The Fc region-dependent actions of IVIG antibodies rely on the quantity of Fc regions that may be wired to self-dimerizing F(ab’)_2_ regions or some other, non-pathogenic targeting capacity. One proposed mechanism is the competition with pAAbs for Fc receptors, but Fc regions of the IVIG antibodies can also prime the immunosuppressive capacity of dendritic cells through FcγRIII-dependent signaling pathways. Alternatively, self-dimerizing IVIG can effectively block effector cell activation. IVIG infusion can also activate regulatory T cells while downregulating T helper 17 cell-dependent immune response [104,105].

Despite the many proposed mechanisms and supportive research, the exact mechanism of action is yet to be defined. Nevertheless, it is evident that the nAAbs have an essential role in regulating immune responses and maintaining a healthy autoimmunity. As part of the whole picture, however, certain IgG nAAbs seem to indicate a loss of immune tolerance, potentially leading to pathological class-switched autoantibodies if affinity maturation occurs [70,101].

## 3. Regulatory Role of the Natural Immune System in Pathological Conditions

### 3.1. From Clonal Selection to Self-Assessment: The Development of Autoreactivity in Immunology

Autoreactivity in healthy individuals has been recognized since the early 20th century, with foundational observations by Besredka in 1901 [106] and Landsteiner (1945) [107] noting the presence of self-reactive antibodies. Since the clonal selection theory from 1959 by Burnet [108] suggested a strong link between autoreactivity and disease, this concept has largely shaped immunological perspectives. However, in 1974, Jerne [109] demonstrated that autoreactivity can exist independently of autoimmune disease and is, in fact, a normal aspect of immune function. Building on this idea, Stewart (1992) [110] hypothesized that natural antibodies evolved primarily to recognize self, with non-self-recognition emerging later in evolution. Avrameas and colleagues further described the immune system as an “extraordinary tool for self-assessment”, emphasizing its role in physiological autoreactivity [111].

Irun Cohen [96] specified that the immune system is composed of networks of interacting cells and molecules, and therefore, we need to apply the thinking and tools of systems immunology to understand and regulate immune system behavior. He defined the HSP60 and HSP70 molecules as examples of key hubs in physiological regulatory networks. HSP molecules, similar to other genetically highly conserved proteins and peptides, can be considered natural system controllers, e.g., to modulate inflammatory responses. Irun Cohen termed this natural autoimmune structuring of the immune system the immunological homunculus—the immune system’s representation of the body. It is a selective advantage of an immune system expressing patterns of built-in autoimmunity to particular sets of self-molecules, suggesting that the particular self-reactivities comprising the homunculus could serve as a set of biomarkers that help the immune system initiate and regulate the inflammatory processes that maintain the body [1,62].

### 3.2. Shifting Balance Between Physiological and Pathological Autoimmunity

Despite the scientific achievements detailed above, the boundary between physiological autoreactivity and pathological autoimmunity remains unclear [111,112,113,114,115]. Interpretation of the first observations suggested that tight regulation limits isotype switching and prevents somatic mutation in B-1 cells to avoid high-affinity IgG autoantibodies that could lead to autoimmunity [68,71,112]. Later studies on human samples, including responses to old (e.g., MMR vaccine) and new (SARS-CoV-2) antigens, showed that nAAb levels can change during immune activation, especially in the IgG type. Changes in nAAb levels in SLE also indicate flexible immune regulation. For instance, anti-dsDNA IgG-positive SLE patients exhibited elevated natural IgG antibodies to specific antigens, resembling an adaptive immune response, and fluctuations in nAAb levels in SLE also pointed to dynamic immune regulation [100,116,117,118].

Experiments in mouse models indicated the necessity of developing bioinformatic tools to study the human nAAb repertoire.

It has been found that human nAAbs are organized into clusters that can distinguish healthy individuals from patients with, e.g., type 1 diabetes mellitus, type 2 diabetes mellitus, or Behçet’s disease [119].

### 3.3. Challenging Conventional Views: Natural Autoantibodies and Their Dynamic Responses in Health and Disease

For decades, the prevailing view of natural autoantibodies has centered on their stable and conserved nature, with minimal fluctuation or adaptive variation across time, sex, and individuals. In seminal work 30 years ago, Coutinho (1995) [117] described natural autoantibodies of the IgM, IgG, and IgA classes as universally present in normal individuals, with reactivity to various serum proteins, cell surfaces, and intracellular structures. These antibodies are even found in human umbilical cord blood and in “antigen-free” mice, with their variable region repertoire shaped by the body’s antigenic landscape and conserved throughout life. Encoded by germline genes with few or no mutations, natural autoantibodies are inherently multireactive and typically lack affinity maturation in healthy individuals. This conserved nature allows natural autoantibodies to contribute broadly to physiological functions, including immune regulation, homeostasis, repertoire selection, resistance to infection, and transport and modulation of biologically active molecules [117].

Challenged by the above ideas, Czömpöly and Nemeth in 2006 investigated whether anti-mitochondrial citrate synthase autoantibodies are components of the natural antibody network in humans. For IgM nAAbs, they found that natural IgM autoantibodies to citrate synthase (CS) are present from infancy, remain stable in adults, may serve as a first line of defense against pathogens, and exhibit unique epitope recognition patterns in pathological conditions such as systemic lupus erythematosus (SLE), suggesting a potential link between innate immunity and autoimmune processes [97,120]. Due to the limited mutation and self-sustained constant presence of the source B cell population (B-1 cells), nAAbs were considered constant. Another study in rodents in 2006 investigated the relationship between the natural antibody repertoire and the host biome. The results showed that habitat (wild vs. laboratory) had a greater effect on immunoglobulin levels than age, strain, or sex. Wild rodents exhibited heightened immune responses similar to autoimmune (Th1-IgG) and allergic (Th2-IgE) responses with putative protective properties, challenging the notion of nAAb constancy, at least with respect to class-switched isotypes [121].

### 3.4. Natural Autoantibodies in Health and Disease: Interplay Between Immunological Response and Pathogenesis

In 2008, researchers “dissected” the cryoglobulins present in hepatitis C (HCV) infection and identified IgM nAAbs targeting anti-HCV IgG1/Κ Fab (VH1-69) that expanded upon infection and contributed to cryoglobulinemia. This study demonstrated the potential characteristics of IgM nAAbs: to expand upon immunological events and to cause indirect damage in non-physiological circumstances, with implications for HCV pathogenesis [122]. However, conflicting results complicate the understanding of nAAbs; a 2010 study [123], also using human samples, focused on IgM anti-Hsp60 levels as a known risk marker in atherosclerosis. Results showed that IgM anti-Hsp60 levels remained stable over a 5-year period, supporting the hypothesis that the immune system selectively preserves autoreactive B cells that target key self-antigens, including Hsp60. The study also showed that anti-HSP60 IgM nAAb levels did not correlate with maternal levels, indicating that nAAb patterns are independent of parental inheritance and specific to the fetus. Regarding IgG nAAbs, a study [90] showed that IgG nAAb diversity in human serum increases with age and is generally higher in women than in men, while certain diseases such as Alzheimer’s, Parkinson’s, and multiple sclerosis are associated with fewer detectable autoantibodies, from which some can potentially be specific to the disease and possibly reflect disease-related immune modulation, highlighting their potential in biomarker and therapy-target research [124,125,126,127,128].

Recently, anti-cytokines (ACAAs) regained attention due to the surge of COVID-19 pandemic-derived data. Anti-type I IFNs have been shown to be risk factors in life-threatening COVID-19 [129], and anti–IL-23 antibodies are linked to adult-onset immunodeficiency [130]. However, antibodies targeting cytokine storm components (TNF-α, IL-1β, IL-6) are in clinical trials [131], justifying the question of whether these antibodies against these targets, if they are naturally present, would provide resilience against pathogen-induced overreactions.

From the perspective of autoimmunity, in RA, anti-IL-1α and anti-TNF levels imply a slower disease progression [132,133]. Anti-INF-α was presented as a mitigator in SLE [134,135], while anti-INF-γ correlates with higher severity [136].

Multiple articles conclude that ontogeny and the clinical meaning of ACAA levels vary strongly by cytokine target and context of investigation [134,135,136].

A review of ACAAs by Schrader and Goding (2014) [137] discusses anti-type I/II interferon, anti-IL-1α, anti-TNF, anti-IL6, anti-GL-CSF, and other ACAAs in great detail. A high affinity and/or high concentration of antibodies against cytokines are considered pathogenic (therefore deemed pAAbs), as they inhibit the physiological action of cytokines, while antibodies measured from healthy human serum (considered nAAbs) tend to have limited affinity and are believed to act as a buffer, reducing the peak concentration of free cytokines and prolonging half-life [137,138]. Plasma levels of IL-1α-, IL-6-, IL-10-, IFNα-, and GM-CSF-specific autoantibodies are present in healthy individuals [134,135]. Their prevalence was 86% of 8972 healthy blood donors, indicating ACAAs as a widespread phenomenon [134].

Overall, natural ACAAs (as nAAbs) can be considered components of an individual’s own immunophenotype [135], which come with both health benefits [131,132,133,135,136,139] and risks [129,140]. Understanding and exploring the realm of ACAAs is very much in line with the aspirations of personalized medicine.

### 3.5. From BCG and SARS-CoV-2 to Natural Autoantibodies: Investigating the Non-Specific Immune Enhancements and Their Mechanisms

Human observations of non-specific effects (NSEs) of the Bacille Calmette–Guérin (BCG) vaccine suggest the involvement of both adaptive and innate immune mechanisms in trained immunity. This memory-like property of innate immune cells results from epigenetic reprogramming after exposure to a primary stimulus such as BCG, which subsequently affects cytokine production and cell metabolism [141]. Understanding the NSEs of vaccination may help to improve the efficacy and safety of future vaccines. NSEs may arise from trained innate immunity, emergency granulopoiesis, or heterologous T cell immunity [142]. Studies investigating the behavior of nAAbs in response to immunization have suggested, initially in rats, that immunization with different allergens enhances natural antibody networks (with a more pronounced effect on IgM than on IgG) [91].

Investigation of a fatal COVID-19 case (2022) showed that de novo natural IgM λ-antibodies can emerge targeting the M antigen of the MNS blood group on RBCs without cross-reacting with SARS-CoV-2 antigens. This first report of a natural bystander anti-RBC antibody highlights the extrafollicular humoral response in severe COVID-19 [143]. Böröcz et al. conducted human serological studies in 2023 to explore the role of nAAbs in adaptive immunity and NSE [144], observing a statistically significant positive association between anti-HSP60, anti-HSP70, and anti-CS IgG titers and anti-SARS-CoV-2 IgG-positive serum levels, especially in mRNA vaccine recipients. Elevated anti-CS IgM levels were also found in samples with a good response to vaccination (indicated by positivity for anti-SARS-CoV-2 IgG, IgA, and IFN-γ). The continuation of this study showed that IgM nAAb levels are significantly related to anti-viral IgG autoantibody levels of “old” immunization (MMR vaccination/infection), and IgG nAAb levels are related to recently established anti-viral (anti-SARS-CoV-2) antibodies [145].

### 3.6. Natural Autoantibodies as Biomarkers and Modulators in Autoimmune Disorders: From Systemic Sclerosis Through Type-1 Diabetes to Hashimoto Thyroiditis in Pregnancy

Czömpöly et al. reported the presence of both pathological autoantibodies and nAAbs on the DNA topoisomerase I molecule in specific different epitopes at the same time [146]. The N-terminal domain-specific pathological (IgG) autoantibodies were predominantly present in the diffuse cutaneous form of systemic sclerosis, while the nAAbs against the F4 fragment were also present in healthy individuals. Moreover, it was found that anti-CS IgG antibodies were significantly increased in anti-dsDNA IgG-positive compared to anti-dsDNA IgG-negative SLE patients [147]. The levels of anti-F4 and anti-CS IgM natural antibodies were significantly increased in anti-dsDNA IgM-positive compared to anti-dsDNA IgM-negative SLE patients. The study also considered the association of nAAbs with virus-induced antibodies in SLE and found significantly higher levels of anti-CS IgG in anti-measles IgG-seropositive samples compared to seronegative samples in rheumatoid arthritis (RA), SLE, and systemic sclerosis (SSc). A subsequent study [148] provided additional insight into the association of IgG anti-CS nAAbs with active SSc, which may indicate compensatory immune responses that fail to counteract disease progression, highlighting their potential as complementary biomarkers alongside double-negative 1 (DN1) B cell ratios for the assessment of disease activity in SSc. Progress in exploring the role of B-1 B cells and nAAbs in type 1 diabetes (T1D) has shown that B-1 B cell-derived N-acetylglucosamine-specific IgM binds β cell antigens, suppresses diabetogenic T cells, and delays T1D in recipients, suggesting a protective role in T1D [149].

Recent results from 2024 include a study showing a negative correlation between serum natural autoantibodies (CS IgM) and complement component C3 in diffuse cutaneous SSc, suggesting that natural autoantibodies may trigger C3 activation and hence consumption, potentially leading to tissue damage [150]; a serological follow-up study in pregnant women with Hashimoto’s thyroiditis [150]; and a serological follow-up study of pregnancies in Hashimoto’s patients and healthy individuals [151], which showed that pregnant women with Hashimoto’s thyroiditis have elevated levels of anti-Hsp60 and anti-Hsp70 IgM nAAbs from the first trimester onwards, accompanied by lower levels of anti-Hsp70 and Hsp60 IgG nAAbs in the third trimester, suggesting a compensatory mechanism that may contribute to maternal immunological tolerance towards the fetus.

Multiple nAAbs have emerged as potential biomarkers, some of which are listed in Table 1 as examples that may help elucidate physiological and pathological autoreactivity and be helpful in the diagnostics or prognostics of autoimmune disease.

### 3.7. Antibodies Against Complex Self-Patterns: The Case of AMPAs and aPls

Physiological and pathological processes involve enzymatic or spontaneous chemical alterations of proteins, exemplified by citrullination, carbamylation, acetylation, glycation, or oxidation, that result in post-translationally modified (PTM) peptide structures that create neo-epitopes or reveal cryptic epitopes that are consequently targeted by anti-modified protein antibodies (AMPAs). Anti-citrullinated protein antibodies (ACPAs) are a family that includes antibodies against citrullinated pathogenic proteins such as Epstein–Barr virus nuclear antigen 1 and autoantigens including keratin, filaggrin and filaggrin-derived peptides (e.g., CCPs), vimentin, and fibrinogen. The 2010 ACR-EULAR classification criteria for rheumatoid arthritis include serological testing of ACPA autoreactivity [157]. AMPAs appear as germ-line-coded nAAbs, and a 2021 study showed that limited mutations are enough to achieve broad-spectrum anti-modified protein reactivity with IgM but not with IgG isotype [158]. The same research suggests that the breakdown of tolerance in rheumatoid arthritis can happen before B cells undergo somatic hypermutations, and AMPA IgM can activate the complement system, participating in synovitis. This might exemplify when “natural” is not always equal to “healthy” and can be a showcase of how even a limited alteration in the germ-line genes can shift nAAbs from physiological to pathological contributors. Whether mutations are the result of chronic activation [98] or due to other reasons remains unclear.

Interactions of phospholipids with binding proteins such as β2-glycoprotein I can expose cryptic epitopes or generate neoepitopes, then these may be targeted by anti-phospholipid (aPls) antibodies. In the case of thrombotic and obstetrically complicating anti-phospholipid syndrome (APS), persistent (measured at least 12 weeks apart) lupus anticoagulant (LAC), anti-cardiolipin (aCL), and aβ2-GP I autoantibodies are part of the 2023 ACR/EULAR anti-phospholipid syndrome classification criteria [159]. This leaves clinicians with 10 to 30% of non-criteria or “seronegative” APS (SN-APS) [160], where emerging literature identified further biomarkers that may hold both pathogenic relevance and diagnostic utility, such as the anti-phosphatidylserine/prothrombin complex (aPS/PT) IgG, anti-vimentin/cardiolipin complex (aVim/CL) IgG, and anti-carbamylated-β2-glycoprotein I (aCarb-β2-GPI) IgG, and aβ2-GPI-domain 1 [161].

Cabiedes et al. (2002) characterized IgM nAAbs against phosphatidylcholine, which also cross-reacted with cardiolipin and vimentin, as well as with dsDNA, ssDNA, and aggregated gamma globulin [162]. Since then, growing evidence suggests that anti-phospholipid antibodies, as IgM nAAbs, have a physiological role [163,164,165]; however, mouse models show that rapidly induced IgG nAAb aPl is pathological [166,167]. Interestingly, in mice, induction of ACPA or anti-carbamylated protein antibody results in the occurrence of other AMPAs in a dynamic cascade manner, highlighting that an AMPA found at diagnostics is not necessarily the first perpetuator of the disease [168].

The biochemistry in autophagosomes favors protein modification, as demonstrated with fibroblast-like synoviocytes from RA patients [169,170], while many of these modified proteins can be found inside or superficially attached to the extracellular vesicles secreted as the final act of autophagy, since the exosome–autophagy network is tightly related [171].

To summarize, contemporary information about natural and pathological aPls and AMPAs and the generation of complex self-patterns (phospholipid-bound and PTM proteins) allows us to speculate that the breakdown of tolerance happens:Very early in the pre-clinical phase [158,172],By either:∘Accumulating mutations on initial germ-line genes of nAAbs [158,162,168], shifting their physiological roles into pathological involvement,∘Or by overcoming tolerance mechanisms of the adaptive immune system due to excessive generation and/or inept clearance of PTM proteins while picturing nAAbs as “innocent” [163,164,165].Both processes are hallmarked by chronic activation [3].Both processes are also fueled by autophagy via the exosome–autophagy network [169,170,171],∘Which is known to be upregulated in response to homeostatic imbalance, environmental stress, and infections [173],∘As well as induced alongside apoptosis [173].

Altogether, this presents the risk of locking the immune system into the escalating vicious cycle of targeting, disrupting, and re-targeting (by epitope-spreading) self-patterns.

## 4. Concluding Remarks

The historical timeline of research into natural autoantibodies (nAAbs) reveals a dynamic evolution in our understanding of their role in the immune system (Appendix A). Initially characterized by their conserved and stable nature, nAAbs were long regarded as mere byproducts of immune

e function, providing a baseline reactivity to self-antigens without significant variation over time or between individuals. Early studies demonstrated their presence in different immunoglobulin classes and highlighted their multireactive nature, suggesting a fundamental role in immune regulation and homeostasis.

As research progressed, the recognition of nAAbs as more than static entities began to take shape. Investigations revealed their ability to adapt in response to infectious agents and vaccination, suggesting a modifying role in bridging innate and adaptive immunity. Studies showed that nAAbs could expand in specific contexts, indicating that they could serve as first-line defenders against pathogens while also contributing to broader immune responses. This new understanding paved the way for exploring their implications in autoimmunity, where the line between protective and pathogenic roles has become increasingly blurred.

Recent research has highlighted the role of natural autoantibodies (nAAbs) as modulators or influencers in various autoimmune diseases, highlighting their importance in conditions such as systemic lupus erythematosus, rheumatoid arthritis, and systemic sclerosis. In addition, their involvement in processes such as immunological tolerance during pregnancy further emphasizes their complexity and vital importance in both health and disease.

In conclusion, the path from the recognition of the conserved nature of natural autoantibodies to the understanding of their multiple roles shows a remarkable shift in immunological perspectives. This evolving narrative highlights the need for continued research to unravel the complex relationships between nAAbs, autoimmunity, infection, and vaccination, ultimately improving our understanding of immune system dynamics and informing therapeutic strategies. The study of nAAbs not only enriches our knowledge of immune function but also opens new avenues for exploiting their potential in clinical applications. All this justifies once again treating natural immunity as a distinct compartment of the immune system, which carries the properties of both innate and acquired immunity in a functional network.

## 5. Implications of the Study

The study of nAAbs provides important insights into their role in immunity and autoimmunity, but several biases and implications must be acknowledged. In particular, this review could not encompass the extensive and heterogeneous scientific literature of the last 40 years, which often presents conflicting results. Variability in experimental designs, methodologies, and populations may contribute to these discrepancies, potentially leading to biased interpretations of the functions of nAAbs.

In addition, the complexity of nAAbs, which can act as both protective and pathogenic factors, complicates their analysis, as their roles can vary significantly depending on the context, including specific autoimmune diseases and individual immune histories. Consequently, the conclusions drawn may not fully capture the diverse nature of nAAbs.

In conclusion, although this study advances our understanding of nAAbs, its results should be interpreted with caution. Future research should aim for a more integrated approach that reconciles the conflicting evidence, ultimately improving our understanding of the importance of nAAbs in health and disease and informing therapeutic strategies.

## Figures and Tables

**Figure 1 ijms-26-05413-f001:**
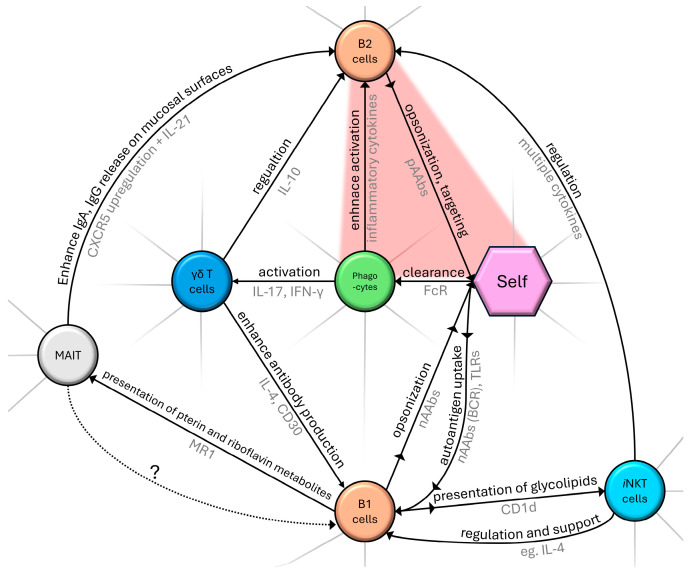
Schematic diagram summarizing a segment of the complex network of cellular components and their role in the natural immune system. Black text marks the nature of interaction, with an arrowhead indicating the direction of effect. Gray text represents examples of interacting molecular pathways. Dotted arrow (with ‘?’ label) indicates our proposed connection between MAIT and B1 cells as described in Section 2.3. The red area indicates pathological self-targeting. Without claiming to be complete, only routes that were mentioned in our review are shown.

**Figure 2 ijms-26-05413-f002:**
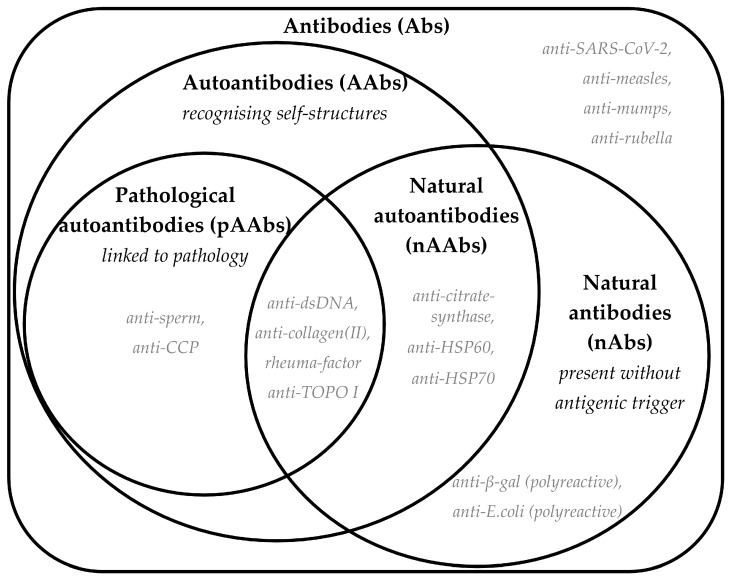
A Venn diagram showing definitions of antibody classification according to their origin and target. Bold text shows the terminology and abbreviations, and definitions are provided in italics. Each set shows a few potential examples with grey italic text; however, it is worth noting that the literature has limited information on the exact mechanism of generation of these antibodies and the links to disease or health.

**Table 1 ijms-26-05413-t001:** Table summarizing nAAbs as potential biomarkers in autoimmune conditions.

Category	nAAb Type/Target	Disease Context ^1^	Key Insight	Source
Protective IgM nAAbs	IgM against apoptotic cells, oxidized self-antigens	SLE	Higher IgM levels are linked to milder disease and protection from nephritis and atherosclerosis.	[74]
Prognostic marker	IgM anti-topoisomerase I (Topo I)	SSc	The presence of IgM anti-Topo I alongside IgG anti-Topo I predicts more rapid skin/lung progression.	[146,152]
Inflammation correlate	IgM/IgG nAAbs against mitochondrial enzymes, HSPs	RA, AS	Baseline and therapy-modulated levels predict CRP, disease activity, and vascular health.	[153]
Disease risk marker	Autoantibodies (e.g., anti-type I IFNs)	T1D, APS-1	The presence of neutralizing IFN autoantibodies correlated with protection from type 1 diabetes.	[154]
Early biomarkers	IgM nAAb “fingerprints” (from arrays)	T1D, RA, SLE	Autoantibody profiles may distinguish preclinical patients and predict autoimmune progression.	[116,155,156]

^1^ The disease context of findings—systemic lupus erythematosus (SLE), systemic sclerosis (SSc), rheumatoid arthritis (RA), ankylosing spondylitis (AS), type 1 diabetes (T1D), and autoimmune polyendocrine syndrome type 1 (AIRE-deficiency; APS-1)—labelled accordingly.

## Data Availability

The authors declare that no data were generated and submitted and hence not applicable.

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
