# Peer review of "Regulators and Conductors of Immunity: Natural Immune System in Health and Autoimmunity"

_ijms, 2025, doi:10.3390/ijms26115413_

Round 1

Reviewer 1 Report

Comments and Suggestions for Authors

An excellent review by Prof Peter Nemeth et al which is an expert in the field of natural autoantibodies (nAAbs). The manuscript is well written with more than 150+ reference, and an Appendix summarising some of the key findings.

For the Title, would “Regulators” be more appropriate than “Guardians”?

Suggestions:

  1. While the review does discuss the involvement of nAAbs in various diseases, it could strengthen the clinical relevance by focusing further on the potential of nAAbs as biomarkers for diagnosis or prognosis in Section 3.6 (line 368-399).
  2. A brief description of anti-cytokine nAAbs in immunodeficiency would be informative, ie, anti-IFNgamma nAAbs in the context of Covid-19.
  3. If possible, a figure illustrating the interactions between γδ T cells, iNKTs, MAIT cells, and B-1 cells would be helpful to those with little immunology background.
  4. This reviewer believe a more detail mechanistic discussion on IVIG nAAbs would be beneficial as the authors rightly point out it is employ for the treatment of MG & RA (line 231-235), but often in other difficult to manage rheumatological conditions from SLE to dermatomyositis.

Author Response

Response to Reviewer Comments
Manuscript title: Regulators and Conductors of Immunity: Natural Immune System in Health and Autoimmunity

Dear Reviewers and Editorial Team,

We would like to express our sincere gratitude for the valuable time, expertise, and constructive feedback provided during the review process. We are immensely grateful for the insightful comments and thoughtful suggestions, which have greatly contributed to improving the clarity, depth, and overall quality of our manuscript.

We genuinely believe that by carefully following the reviewer guidelines and incorporating the recommended changes, our article has become significantly stronger. The scientific value of the manuscript has increased exponentially as a result of these revisions.

Please be assured that all the requested modifications have been thoroughly elaborated and fully integrated into the revised version of the text. We have addressed each point in detail and made the corresponding changes throughout the manuscript where appropriate.

Once again, we thank you for your valuable input and consideration, and we remain hopeful that the revised version meets your expectations.

Sincerely,

Katalin Böröcz

Reviewer 2 Report

Comments and Suggestions for Authors

This comprehensive review offers a valuable synthesis of the role of natural autoantibodies and cellular components of the natural immune system in both health and disease. The manuscript presents an in-depth and timely discussion of natural immunity, drawing on recent advances and integrating decades of literature. It effectively bridges innate and adaptive immune perspectives, while also highlighting the dual protective and pathogenic roles of natural autoantibodies.

Although in my opinion the manuscript requires only minor revisions, I would appreciate it if the authors could consider antiphospholipid antibody syndrome. In particular, some antigens—such as the recently described carbamylated beta-2 glycoprotein I—lead to the formation of autoantibodies. Moreover, a very recent study published in Clinical and Experimental Immunology in 2024 reported on the various forms of autoantibodies found in patients with “seronegative” antiphospholipid syndrome.

Recent studies (particularly in 2023) have reviewed the role of post-translationally modified antigens in rheumatoid arthritis , particularly those carried by extracellular vesicles or generated through autophagy. Could these mechanisms represent a bridge between physiological autoreactivity  and the development of pathogenic autoimmunity?

Author Response

(The authors gave the same response as above.)
